# Negative Outcome of Temporalis Fascia Graft in Tympanoplasty with Excessive Bleeding: A Retrospective Study

**DOI:** 10.3390/medicina59010161

**Published:** 2023-01-13

**Authors:** Andrea Lovato, Antonio Frisina, Andrea Frosolini, Daniele Monzani, Roberto Saetti

**Affiliations:** 1Department of Neuroscience DNS, Audiology Unit at Treviso Hospital, University of Padova, 31100 Treviso, Italy; 2Otorhinolaryngology Unit, Department of Surgical Specialties, Vicenza Civil Hospital, 36100 Vicenza, Italy; 3Maxillofacial Surgery Unit, Department of Medical Biotechnologies, University of Siena, 53100 Siena, Italy; 4Otorhinolaryngology Unit, Department of Surgical Specialties, University of Verona, 37100 Verona, Italy

**Keywords:** tympanoplasty, graft material, xenograft, temporalis fascia, intraoperative bleeding, tympanic membrane perforation

## Abstract

*Background and Objectives*: Non-autologous graft materials hold promise for tympanic membrane (TM) perforation closure. In the present manuscript, we aimed to evaluate the influence of clinical and surgical (i.e., graft materials) characteristics on tympanoplasty outcome in chronic otitis media (COM). *Materials and Methods*: We retrospectively reviewed clinical and surgical characteristics of COM patients with TM perforation treated with tympanoplasty and mastoidectomy. Univariate and multivariate appropriate tests were applied. *Results*: We used xenograft (porcine submucosal collagen) in 163 patients, and temporalis fascia in 210. The mean follow-up time was 37.2 months. Postoperative TM perforation (i.e., negative outcome) was detected in 11.6% of cases with xenograft, and in 12.8% with temporalis fascia. Performing uni- and multivariate analysis, we determined that large (three or all quadrants) TM perforation (*p* = 0.04) and moderate-to-severe intraoperative bleeding (*p* = 0.03) were independent prognostic factors of negative outcome. Considering the 197 patients with moderate-to-severe intraoperative bleeding, we disclosed that the use of temporalis fascia (*p* = 0.03) was an independent risk factor of postoperative TM perforation. *Conclusions*: According to our results, large TM perforation and moderate-to-severe intraoperative bleeding were independent prognostic factors of negative outcome in adult COM patients treated with tympanoplasty. In the sub-group of COM patients with excessive intraoperative bleeding, use of temporalis fascia was associated with negative outcome; these patients could benefit from xenograft materials. These findings should be tested in large randomized clinical trials.

## 1. Introduction

Outcome of tympanoplasty is a topic of interest in otolaryngology; the influence of specific factors such as age, size of tympanic membrane (TM) perforation, and surgical technique on the outcome remains controversial in the literature [1,2]. Several graft materials have been developed for closure of TM perforation. Traditionally, temporalis fascia and tragal cartilage have been used, together with other autologous materials such as fascia lata, fat, vein, and perichondrium [3]. Although autograft materials are generally of low cost and readily available, they can be associated with donor site morbidity [4]. Furthermore, autologous graft harvest does not always leave sufficient high-quality tissue for revision and second-look surgeries [4]. Unfortunately, tympanoplasty with autologous grafts showed variable outcomes, with several large-scale case series reporting failure rates of up to 20% [1,5,6]. Although poor outcome may be due to surgical technique, ongoing chronic otitis media (COM), persistent eustachian tube dysfunction, and/or ossicular abnormalities, it is important to consider the ways in which graft materials contribute to healing and hearing outcomes. More recently, autograft alternatives have been increasingly employed in tympanoplasty [7,8]. Non-autologous grafts obviate the need for an external incision to harvest graft materials, resulting in a reduction in donor site morbidity, as well as in overall operative time. Furthermore, a superior coverage for large perforations could be expected [9].

In this retrospective study, we evaluated a series of adult COM patients treated with tympanoplasty. Our aim was to evaluate the ways in which clinical and surgical (i.e., graft materials) characteristics could influence the outcome of tympanoplasty in terms of postoperative TM perforations.

## 2. Materials and Methods

### 2.1. Patients and Procedures

The study was conducted in accordance with the principles of the Helsinki Declaration and was approved by the internal Ethical Committee. Data were examined in agreement with the Italian privacy and sensible data laws. Consecutive COM patients treated with canal wall up tympanoplasty at Otorhinolaryngology Unit of Vicenza Civil Hospital (Italy) were enrolled in this retrospective study. The inclusion criteria were the following: (i) COM with TM perforation; (ii) mastoidectomy performed together with tympanoplasty; (iii) age > 18 years old. Patients with craniofacial dysmorphisms, cholesteatoma, ossiculoplasty, or revision surgery were excluded.

Demographic and clinical characteristics were recorded. At first and follow-up visits, all patients underwent otomicroscopy and pure tone audiometry in a silent cabin, including bone-conduction (BC) thresholds at 0.5, 1, 2, and 4 kHz, and air-conduction (AC) thresholds at 0.25, 0.5, 1, 2, 4, and 8 kHz, for both ears (operated and contralateral) [10]. When necessary, in order to eliminate the interaction of the contralateral ear caused by interaural attenuation, a narrow band noise was administered to the better ear while testing the worst [11]. Air-bone gap (ABG) was calculated as the average difference between the air- and bone-conduction thresholds at 0.5, 1, 2, and 4 kHz. Only AC and BC results that were obtained at the same time were used for calculations, according to American Academy of Otolaryngology Head and Neck Surgery guidelines [12]. TM perforation was classified according to size (one, two, three, or all quadrants), as previously reported [13]. Otorrhea was classified according to Bellucci [14] as (i) dry; (ii) occasionally wet; or (iii) persistently wet.

Tympanoplasty and concurrent mastoidectomy was performed with a postauricular approach using the microscope in all patients. A knife was used to scratch the edges of the perforation circumferentially. The tympanomeatal flap was elevated. The tympanic cavity was visualized, and ossicles condition was surveyed. If pathologic lesions were detected (e.g., tympanosclerotic lesions, granulation, or fibrosis), they were removed to mobilize the ossicles. We excluded the cases with poor ossicle mobility in which ossiculoplasty was a better indication. The TM perforation was closed with a medial-to-malleus underlay technique. Different graft types were used: allograft pericardium (Tutoplast; ENTrigue Biologics, San Antonio, TX, USA), xenograft (porcine submucosal collagen; Biodesign; Cook Medical Inc., Bloomington, IN, USA), and autografts (dry temporalis fascia or tragal cartilage). Four surgeons operated on the patients, and every surgeon chose the graft type independently, according to their own evaluation and preference. Graft size was two to three times larger than TM perforation. Intraoperative bleeding was classified as low, moderate, or severe. Intraoperative complications were recorded.

The general follow-up schedule (adjustable to patient’s individual characteristics) was as follows: (i) every 15 days in the first month after surgery; (ii) once every month in the second and third month; (iii) every 6 months thereafter. We recorded postoperative complications. Postoperative infection was defined as an infection resulting in a patient being prescribed an antibiotic during the first month of follow-up. For postoperative ABG, we considered the value at last follow-up visit.

As outcome variable, we considered postoperative perforation. In postoperative TM perforation, we included persistent perforation (a perforation that did not heal within 3 months after surgery) and recurrent perforation (patients who had documented closure and a subsequent perforation were identified).

### 2.2. Statistical Analysis

We used the Fisher exact test, the Mann–Whitney U test, and the chi-square test as appropriate. For every significant association disclosed by the Fisher exact test, we calculated an odds ratio (OR). When necessary, continuous variables were dichotomized according to the median value, as previously reported [15]. A multivariate logistic model was constructed, adding only the clinical parameters with a *p* value ≤ 0.1, as disclosed by Fisher exact test at univariate analysis according to previous multivariate modelling experience [15,16]. The results were expressed as ORs, *p* values, and 95% confidence intervals (CIs). During the analysis, the model was checked for multicollinearity with a variance inflation factor test. A *p* value < 0.05 was considered significant. The quality of the model was assessed with Pearson chi-squared test, a non-significant result (*p* ≥ 0.05) meaning a good fit of the model to the data. The Social Sciences version 17 statistical package (SPSS Inc., Chicago, IL, USA) was used for all analyses.

## 3. Results

In the present retrospective study, we identified 429 consecutive COM patients that met inclusion criteria. In the tympanoplasty cases, the graft types were allograft pericardium in 24 patients, xenograft in 163, temporalis fascia in 210, and tragal cartilage in 32 patients. We decided to exclude 56 patients in which allograft and cartilage were used in order to consider large homogeneous samples (163 xenograft and 210 temporalis fascia patients).

In Table 1, we reported demographic and clinical characteristics, surgical complications, and the outcome of patients divided in two groups according to graft type. The mean follow-up time was 6.8 years (standard deviation 3.2 years), with no difference between the two groups (*p* = 0.22; Mann–Whitney U test). For comorbidities, in statistical analysis, we considered diabetes (diagnosed in 23 patients), and primary/secondary immunodeficiency (diagnosed in 16 patients), as these pathologies could have influenced the graft healing and facilitated postoperative infections. Intraoperative complications were bleeding from sigmoid sinus in 11 cases, and otoliquorrhea in 1 case (all of these were treated during mastoidectomy). Postoperative complications were vertigo in 35 patients and infections in 21. We observed no significant difference between the two groups in terms of clinical characteristic and operative complications. Postoperative TM perforation was detected in 12% of patients (11.6% with xenograft and 12.8% with temporalis fascia), but no difference was detected between the two groups. After surgery, ABG significantly improved in both xenograft (*p* < 0.0001, Mann–Whitney U test) and temporalis fascia (*p* < 0.0001, Mann–Whitney U test) groups.

Using Fisher exact test, we performed a univariate analysis with dichotomized variables (Table 2) in order to discover an association with the outcome (i.e., postoperative TM perforation). We discovered a significant association for large (three or all quadrants) TM perforation (*p* = 0.02; OR 5.25) and moderate-to-severe intraoperative bleeding (*p* = 0.01; OR 6.2). Univariate analysis disclosed three characteristics with a *p* value ≤ 0.1 at Fisher exact test: temporalis fascia graft type, large TM perforation, and moderate-to-severe intraoperative bleeding. Using these three variables in the model, no multicollinearity was detected by the variance inflation factor test. Whereas temporalis fascia graft type (OR = 1.63; *p* = 0.23, CI 95% 0.65–3.17) was not associated with the outcome, large TM perforation (OR = 3.46; *p* = 0.04, CI 95% 1.32–10.90), and moderate-to-severe intraoperative bleeding (OR = 4.21; *p* = 0.03, CI 95% 1.46–15.90) were independent prognostic factors of postoperative TM perforation. The result of Pearson chi-squared test (*p* = 0.39) showed a good fitting of the empirical model to real data.

Considering the 197 patients with moderate-to-severe intraoperative bleeding, we detected 26 postoperative TM perforations, 6 in the xenograft and 20 in temporalis fascia group (*p* = 0.01; chi-square test). We decided to perform a sub-group analysis in these 197 patients (Table 3).

Using Fisher exact test, we discovered a significant association with postoperative TM perforation for large TM perforation (*p* = 0.03; OR 4.84) and temporalis fascia graft type (*p* = 0.01; OR 5.92). Univariate analysis disclosed two characteristics with a *p* value ≤ 0.1 at Fisher exact test: temporalis fascia graft type and large TM perforation. Using these two variables in the model, no multicollinearity was detected by the variance inflation factor test. Temporalis fascia graft type (OR = 3.23; *p* = 0.03, CI 95% 1.55–9.11) and large TM perforation (OR = 2.86; *p* = 0.04, CI 95% 1.21–8.92) were independent prognostic factors of postoperative TM perforation. The result of Pearson chi-squared test (*p* = 0.56) showed a good fitting of the empirical model to real data.

## 4. Discussion

Non-autologous graft materials may solve several dilemmas in tympanoplasty by obviating the need for graft harvest, facilitating consistent wound healing, and permitting graft placement in the clinical setting. In a recent systematic review, the authors claimed that porcine submucosa and basic fibroblast growth factor could hold promise for chronic perforation closure [17]. In the present study, we reviewed the outcome of COM patients treated with tympanoplasty. Our aim was to search for clinical or surgical variables of outcome. In order to avoid confounding bias and better evaluate the results of different graft materials, patients with craniofacial dysmorphisms, cholesteatoma, ossiculoplasty, and revision surgery were excluded. In the 373 included patients, we disclosed a significant improvement in ABG after surgery. Postoperative TM perforation was detected in 11.6% of patients operated with the xenograft, and in 12.8% of patients that received temporalis fascia, with no significant difference. Our results were comparable with large series including autologous [18] and non-autologous [17] graft materials. In one investigation considering 404 myringoplasties, the success rate of xenograft was 96% [9].

Performing the uni- and multivariate analysis of outcome, we determined that large TM perforations and moderate-to-severe intraoperative bleeding were independent prognostic factors of postoperative TM perforations. Graft type was not associated with the outcome, as reported by other studies that compared xenograft with autologous materials [9,19]. In our patients, large perforation (three or all quadrants) was associated with negative outcome. Large TM perforation was determined to be a negative prognostic factor in large systematic review and meta-analysis [1,20]. Excessive bleeding may limit surgical field clarity, and thus compromise TM reparation. A previous report on 232 cholesteatoma operated cases detected residual pathology in 45 (19%) ears [21]. Multivariate regression analysis demonstrated that intraoperative bleeding was associated with increased risk of residual cholesteatoma [21].

We performed a sub-group analysis in the 197 patients with moderate-to-severe intraoperative bleeding. In this population, the uni- and multivariate model confirmed large TM perforation as a risk factor; temporalis fascia was another independent prognostic factor associated with postoperative perforation. In patients with excessive intraoperative bleeding, we detected significantly less perforations when xenografts were used. To the best of our knowledge, no previous study considered intraoperative bleeding in the outcome of TM reconstruction. We could hypothesize that porcine submucosal collagen does not adsorb blood and maintains its stiffness. On the contrary, a dry temporalis fascia would rapidly become wet in the case of excessive bleeding. Surgeons usually prefer a dry temporalis fascia because it is easier to lay out in TM reconstruction [22]. The relative thickness and lack of form of wet fascia make it harder to position grafts precisely compared with dried fascia [22].

The main strengths of our investigation were the high number of included patients and the long follow-up period. The main weakness was the retrospective design of the study.

## 5. Conclusions

According to our results, large TM perforation and moderate-to-severe intraoperative bleeding were independent prognostic factors of postoperative perforation in adult COM patients treated with tympanoplasty. In the sub-group of COM patients with excessive intraoperative bleeding, the use of temporalis fascia was associated with negative outcome. In COM cases with excessive operative bleeding, patients could benefit from xenograft materials. These findings should be tested in large randomized clinical trials.

## Figures and Tables

**Table 1 medicina-59-00161-t001:** Clinical/surgical characteristics and the outcome of chronic otitis media patients treated with tympanoplasty.

	Xenograft Group	Temporalis Fascia Group	*p* Value
N°. of patients	163	210	
Age in years (SD)	43.2 (12.3)	48.4 (11.4)	0.83
Sex (*n*°. of female)	90	121	0.72
Comorbidities (*n*°. of patients)	20	19	0.4
TM perforation size (three and all quadrants)	45	62	0.77
Duration of perforation in months (SD)	28.1 (15.2)	30.2 (12.1)	0.34
Preoperative ABG in decibels (SD)	22.3 (6.1)	20.2 (5,3)	0.45
Intraoperative bleeding (moderate and severe)	90	107	0.47
Intraoperative complications	5	7	0.88
Duration of surgery in minutes (SD)	163.2 (27.9)	178.3 (30.1)	0.13
Postoperative complications	24	32	1.00
Postoperative ABG in decibels (SD)	9.2 (2.3)	8.4 (2.6)	0.89
Postoperative TM perforation	19	27	0.88
Follow-up times (years)	6.2 (3.7)	7.3 (3.1)	0.22

Abbreviations: ABG (air-bone gap), N (Number), SD (standard deviation), TM (tympanic membrane).

**Table 2 medicina-59-00161-t002:** Univariate and multivariate analysis of outcome (i.e., postoperative TM perforation) in chronic otitis media patients treated with tympanoplasty.

	Univariate Analysis	Multivariate Analysis
Age > 45 years	0.98	
Female patients	0.63	
Comorbidities	1.0	
TM perforation size (three and all quadrants)	0.02	0.04
Duration of perforation > 28 months	0.45	
Persistently wet ears	0.36	
Preoperative ABG > 20 decibels	0.34	
Temporalis fascia (graft type)	0.09	0.23
Intraoperative bleeding (moderate and severe)	0.01	0.03
Intraoperative complications	1.0	
Postoperative complications	1.0	

Abbreviations: ABG (air-bone gap), TM (tympanic membrane).

**Table 3 medicina-59-00161-t003:** Univariate and multivariate analysis of outcome (i.e., postoperative TM perforation) in chronic otitis media patients showing excessive bleeding during tympanoplasty.

	Univariate Analysis	Multivariate Analysis
Age > 48 years	0.79	
Female patients	0.54	
Comorbidities	1.0	
TM perforation size (three and all quadrants)	0.03	0.04
Duration of perforation > 32 months	0.23	
Persistently wet ears	0.15	
Preoperative ABG > 22 decibels	0.44	
Temporalis fascia (graft type)	0.01	0.03
Intraoperative complications	1.0	
Postoperative complications	1.0	

Abbreviations: ABG (air-bone gap), TM (tympanic membrane).

## Data Availability

Datasheets are available upon request to the corresponding author.

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
