# Peer review of "Negative Outcome of Temporalis Fascia Graft in Tympanoplasty with Excessive Bleeding: A Retrospective Study"

_medicina, 2023, doi:10.3390/medicina59010161_

Round 1

Reviewer 1 Report

This article is a retrospective study of tympanic membrane repair outcome in patients undergoing microscope surgery for chronic otitis media, assessing the success rate and advantages of non-autologous graft materials over temporalis fascia for tympanic membrane repair.

The strength of this paper lies in the longer follow-up and more detailed data recording of cases, and the use of multiple statistical methods to compare the efficacy of two tympanic membrane repair materials in microscopic otitis media surgery.

However, there are still some areas of the article that need to be improved and revised.

The surgery for chronic otitis media in this article focuses on tympanic membrane repair with a microscopic retroauricular incision and patients with craniofacial dysmorphisms, cholesteatoma, ossiculoplasty, or revision surgery are excluded. But this procedure can now be largely replaced by endoscopic surgery, which rarely involves significant bleeding and also mostly uses self-cartilage repair. The application scenario in this article is more limited and it would be more clinically relevant to evaluate tympanic membrane repair in revision surgery or in surgery with more severe ear lesion.

Additional information is needed in this article: the method of identifying graft size; whether the type of temporalis fascia is dry or wet; how long the procedure took; how many cases underwent tympanoplasty alone, how many cases underwent both tympanoplasty and mastoidectomy, whether the type of tympanoplasty was up or down, etc. Whether were there possible differences between the different groups.

The few questions about the statistics: 1) why logistic regression was not used, 2) the inclusion criteria for including several single factors in the multi-factor analysis need to be described in detail in the article.

Are references 11, 15 and 16 of this paper necessary?

References 21 and 22 are suggested to be replaced with more appropriaten references.

Author Response

<The surgery for chronic otitis media in this article focuses on tympanic membrane repair with a microscopic retroauricular incision and patients with craniofacial dysmorphisms, cholesteatoma, ossiculoplasty, or revision surgery are excluded. But this procedure can now be largely replaced by endoscopic surgery, which rarely involves significant bleeding and also mostly uses self-cartilage repair. The application scenario in this article is more limited and it would be more clinically relevant to evaluate tympanic membrane repair in revision surgery or in surgery with more severe ear lesion.>

Thank you for the comment. All patients underwent tympanoplasty together with mastoidectomy for chronic otitis media, as reported in Materials and Methods. Consequently, the use of microscope started during the mastoidectomy was continued for tympanoplasty. At our Institution, the endoscope is used for endoaural myringoplasty. We considered primary surgery and excluded revision in order to avoid other confounding bias and better evaluate the results of different graft materials. We stressed this point in first paragraph of Discussion.

<Additional information is needed in this article: the method of identifying graft size; whether the type of temporalis fascia is dry or wet; how long the procedure took; how many cases underwent tympanoplasty alone, how many cases underwent both tympanoplasty and mastoidectomy, whether the type of tympanoplasty was up or down, etc. Whether were there possible differences between the different groups.>

As suggested we provided additional information. Graft size was two-to-three times larger than tympanic membrane perforation; we added this in Materials and Methods. Temporalis fascia was dry, as already mentioned in the third paragraph of Materials and Methods. The duration of surgery was added in Results, Table 1. All patients underwent mastoidectomy together with tympanoplasty, see inclusion criteria in Materials and Methods. Tympanoplasty was canal-wall-up in all cases, we added this in Material and Methods.

<The few questions about the statistics: 1) why logistic regression was not used, 2) the inclusion criteria for including several single factors in the multi-factor analysis need to be described in detail in the article.>

We used a multivariate logistic model that could be easily adapted to different clinical settings (for instance in rhinology and head and neck oncology, see reference number 15 and 16) and gave solid results. As reported in Materials and Methods, the model was constructed adding only the clinical parameters with a p value ≤ 0.1, as disclosed by Fisher exact test at univariate analysis. Previous experience on multivariate modelling supported this methodology (see reference number 15 and 16).

<Are references 11, 15 and 16 of this paper necessary?>

These references supported our Methodology, in particular the multivariate modeling, see previous comment.

<References 21 and 22 are suggested to be replaced with more appropriaten references.>

We removed these references.

A professional English mother-tongue translator corrected the revised version of the manuscript, see Author contribution.

Reviewer 2 Report

It is indeed a good study which has taken into account large number of patient. I have few queries

1. How did one decide which patient group would get temporals fascia and which group would get the xenograft?

2. Was some sort of uniformity maintained in selecting the size of perforation in each group? This would have an effect on overall graft take-up results

3. Has any patient had side effects/ rejection of xenograft among all the patients operated on?

Author Response

  1. How did one decide which patient group would get temporalis fascia and which group would get the xenograft?

The study was conducted in a single tertiary referral center. Nevertheless, four surgeons operated on the patients, and every surgeon chose independently the graft type according to his own evaluation and preference. We added this point in Materials and Methods.

  1. Was some sort of uniformity maintained in selecting the size of perforation in each group? This would have an effect on overall graft take-up results

No there wasn’t. In primary surgery, there are no limitations to the use of temporalis fascia, even on larger perforations.

  1. Has any patient had side effects/ rejection of xenograft among all the patients operated on?

No there wasn’t. We reported complications in Results.

A professional English mother-tongue translator corrected the revised version of the manuscript, see Author contribution.